# Enhancement of Phytoremediation of Heavy Metal Pollution Using an Intercropping System in Moso Bamboo Forests: Characteristics of Soil Organic Matter and Bacterial Communities

Fangyuan Bian [1,2], Xiaoping Zhang [1,2,*], Qiaoling Li [1,2], Zhiyuan Huang [1,2] and Zheke Zhong [1,2]

1 China National Bamboo Research Center, Key Laboratory of State Forestry and Grassland Administration on Bamboo Forest Ecology and Resource Utilization, Hangzhou 310012, China; bianfangyuan@caf.ac.cn (F.B.); liqiaoling65@163.com (Q.L.); zhiyuanhuang@caf.ac.cn (Z.H.); zhekez@163.com (Z.Z.)
2 National Long-Term Observation and Research Station for Forest Ecosystem in Hangzhou-Jiaxing-Huzhou Plain, Hangzhou 310012, China
* Correspondence: xiaopingzhang@caf.ac.cn; Tel.: +86-0571-88860734; Fax: +86-0571-88860944

**Abstract:** Heavy metal pollution in soil is a major global issue, and one effective method for addressing it is phytoremediation through bamboo planting. Nevertheless, there is a notable gap in our knowledge as no studies have explored the characteristics of soil organic matter (SOM) and the bacterial communities in bamboo forests during the remediation process. To bridge this knowledge gap, we conducted research to investigate the impact of different bamboo planting patterns on the SOM characteristics and microbial communities in soils contaminated with heavy metals. The contents of SOM and dissolved organic matter (DOM) in rhizosphere and non-rhizosphere soils differed significantly between monocropping and intercropping systems, with DOM accounting for only 1.7%–2.5% of SOM. Fourier transform infrared spectra showed that the contents of SOM polysaccharides C-O, carbonate C-O, aliphatic methyl, and methylene increased, while the aromatic C=C abundance decreased in the intercropping rhizosphere soil. The differences between bamboo cultivation patterns in the rhizosphere and non-rhizosphere soils were elucidated using the biomarkers, including *MND1* and *Nitrospira* (non-rhizosphere), and *Sphingomonas* (rhizosphere). Heavy metals, DOM, SOM, and refined organic functional groups, especially C-O in polysaccharides and symmetric carboxylate, were the determining factors of soil bacterial communities. Compared to monocropping, intercropping increased the accumulation of Zn and Cd in bamboo shoots by 35% and 40%, respectively, and hence, intercropping soil, with a low toxicity, was suitable for bamboo shoot sprouting. Intercropping can alter the characteristics of SOM and bacterial communities and plays a vital role in phytoremediation and shoot growth in bamboo forests. Future studies on soil carbon dynamics and nutrient status during heavy metal remediation will improve our knowledge of soil transformation and its impact on soil ecosystem health and productivity.

**Keywords:** bacterial community; bamboo forest; heavy metal pollution; intercropping; phytoremediation; soil organic matter

## 1. Introduction

The rapid development of industrialization during the past century has exacerbated the problem of environmental pollution, with heavy metal pollution emerging as a global concern [1]. Soils in more than 5 million locations worldwide are contaminated with heavy metals. In China, 2.9% of arable land (approximately 40,000 hectares) is moderately or severely polluted, and more than 20 million acres of agricultural land (25% of the total farmland area) is contaminated [2]. Soil heavy metal pollution has been widely researched both domestically and internationally [3,4]. Hu, et al. [5] classified the 31 provinces of China into six subsets and determined that the Zhejiang Province was in the more vulnerable

first subset based on potential driving factors of soil heavy metal pollution, such as a high gross domestic product, number of civilian cars, and abundance of mineral resources. Sites with high heavy metal contamination need strict environmental monitoring to apply remediation measures.

Bamboo is the only major woody grass in forests and is one of the most valuable non-timber forest products. Moso bamboo has a significant potential for the remediation of heavy metal-contaminated soil owing to its high biomass, short rotation cycle, and high economic value [6], and it has been the focus of many recent experiments [7–9]. Furthermore, Moso bamboo captures and sequesters carbon (C) dioxide in the atmosphere, plays an important role in the terrestrial C cycle, offers major ecosystem services, and is ecologically and economically valuable [10]. Global warming promotes the long-term C sequestration by Moso bamboo [11].

Soil organic matter (SOM) is considered a primary factor in enhancing soil heavy metal bioavailability and chemical behavior [12] and is a key parameter that influences soil quality and crop productivity [13–15]. SOM is composed of many functional groups, including carboxyl, phenolic, hydroxyl, and amino groups. These functional groups can inhibit or promote the migration of heavy metals in soils through a variety of interactions, such as adsorption, complexation, and chelation [4]. SOM is classified into rhizosphere SOM and non-rhizosphere SOM based on the distance from the plant roots and has distinct differences in terms of composition and physicochemical characteristics [16]. Therefore, understanding the characteristics of SOM in the rhizosphere and non-rhizosphere soils is beneficial for the phytoremediation of heavy metals.

Previous studies have shown that microbial communities in the rhizosphere can be recruited with specific beneficial functions, including providing mineral nutrients for plants [17], enhancing the plant immune system [18], and helping plants resist heavy metals [19]. Soil microbial communities are indicators of the remediation of contaminated soil as they influence soil pH, SOM, and soil physicochemical properties [20], and in turn, they are significantly influenced by environmental heterogeneity, plant species, and cultivation patterns [3,21]. The relationship between soil environmental factors and soil microorganisms and their correlation with soil heavy metal types and cultivation patterns has rarely been reported. Our previous study demonstrated that bamboo intercropped with *Sedum plumbizincicola* can be an effective system for heavy metal removal from contaminated soil [22] (Table S1). Soil bacteria and properties have major influences on the restoration of bamboo forests [7]; however, the roles of the characteristics of SOM in heavy metal remediation have not been systematically investigated so far. Moso bamboo produces high-quality edible bamboo shoots. Heavy metal distribution in the soil induces their accumulation in the edible parts of the phytoremediation species [1]. Therefore, it is important to understand the link between the growth of new bamboo shoots and the removal rate of heavy metals.

In the present study, the effects of monocropping and intercropping practices in the cultivation of bamboo on the composition of bacterial communities were compared. The differences in the characteristics of organic matter of rhizosphere and non-rhizosphere soils in bamboo forests were investigated. The specific objectives of this study were to (1) determine the soil microecological characteristics of bamboo under different cultivation patterns, including SOM and bacterial community structure, (2) analyze the relationship between soil and microorganisms in the process of phytoremediation, and (3) evaluate the growth characteristics and heavy metal accumulation ability of new bamboo shoots under heavy metal stress.

## 2. Materials and Methods

### 2.1. Study Site and Management

The experiment was conducted in Fuyang (119°25′00″–120°19′30″ E, 29°44′45″–30°11′58.5″ N), Hangzhou of Zhejiang Province, China. The region has a tropical to subtropical monsoon climate characterized by four distinct seasons and abundant rainfall. The region experiences

an annual average temperature of 16 °C, relative humidity of 70%, precipitation of 1452 mm, and sunshine duration of 1709 h. Fuyang is situated on a flat plain with predominantly fine sand and clay surface sediments dominating the landscape. The soil at the research site falls under the classification of Ferralsol, as per the soil classification system established by the Food and Agriculture Organization of the United Nations. This Ferralsol is characterized as red soil and has developed from the parent material of siltstone [23]. The average values for the basic physicochemical properties at the study site were: pH of 7.83; bulk density of 1.66 g/cm$^3$; water content of 210.57 g/kg; total porosity of 42.25%; organic C of 25.8 g/kg; total nitrogen (N) of 2.41 g/kg.

Notably, in proximity to our research site, a hardware electroplating factory had been discharging industrial wastewater into farmland irrigation channels. However, this discharge practice was terminated in 2007. The electroplating factory's wastewater contains a range of heavy metal ions such as Zn, Cu, Cd, Ni, and Pb, as well as other hazardous components, including acids, alkalis, cyanides, and various toxic debris. The soil in the study site was polluted with industrial wastewater, leading to the presence of heavy metals. Preliminary soil analysis indicated that the primary contaminants were Zn, Cu, and Cd. The heavy metal contaminated soil was remediated using monocropping of Moso bamboo and intercropping of Moso bamboo and *Sedum plumbizincicola*. Treatments with three replicates of 10 × 20 m plots were established in a randomized block design. The spacing of the bamboo planting row was 2 × 2.5 m. The size of the planting hole was 60 × 60 × 60 cm, and the mother bamboo was planted in the center of the hole. *S. plumbizincicola* was planted around the bamboo. The spacing between *S. plumbizincicola* and bamboo rows was 20 cm.

### 2.2. Soil Sampling and Physicochemical Properties

After bamboo and *S. plumbizincicola* were planted, soil samples were collected from the study site at five points over a period of 5 years at an average depth of approximately 20 cm. Rhizosphere soil was obtained using the root-shaking method. Non-rhizosphere soil was dug up using a shovel from an area distant from the rhizosphere. Rhizosphere soils of bamboo in the monocropping system (RM) and intercropping system (RI) and non-rhizosphere soils of bamboo in the monocropping system (NRM) and intercropping system (NRI) were placed in a transparent self-sealing bag and stored in an approximate vacuum by manually removing the air. The samples were immediately transported to the laboratory. Samples used for the microbiological experiments were stored at −80 °C. Soil samples used for analyses of physical and chemical properties were stored in a neutral environment, dried, ground, and sieved through a 2-mm sieve.

Triplicates of each soil sample were used to determine SOM and dissolved organic matter (DOM) contents. In addition, triplicate analyses of heavy metal contents of soils were performed. Soil samples were dried and acidified, and SOM was measured using a total organic C analyzer (Multi N/C 3100, Analytik Jena AG, Jena, Germany). DOM was determined using $K_2SO_4$ [24]. After the soil was digested using $HNO_3$–HCl–$HClO_4$, the heavy metal content in the soil was determined using atomic absorption spectrophotometry (AAnalyst800, Perkin Elmer, Waltham, MA, USA) [25]. Bamboo shoots were dried and ashed, and their heavy metal contents were determined using atomic absorption spectrophotometry (AAnalyst800, Perkin Elmer, USA).

### 2.3. High-Throughput Sequencing

Soil microbial DNA was extracted from the rhizosphere and non-rhizosphere soils using a Soil DNA Isolation Kit (Tiangen Biotech Co., Ltd., Beijing, China) according to the manufacturer's instructions. The isolated DNA was stored in a freezer at −20 °C until further analysis. After DNA extraction, polymerase chain reaction (PCR) assays were used to amplify the V3–V4 region of the 16S rRNA gene using the optimized primer pair 341F/805R using respective barcodes. The PCR products were sequenced on the Illumina MiSeq platform (Illumina Inc., San Diego, CA, USA) by G-BIO Technology Co., Ltd. (Hangzhou, China).

### 2.4. Fourier Transform Infrared (FTIR) Spectroscopic Analysis

FTIR spectroscopy was used to analyze the soil surface chemical groups. Soil samples were dried at 70 °C before analysis. One milligram of soil (sieved to <0.5 mm) from each sample was mixed with 100 mg KBr and ground with an agate mortar under the irradiation of a heating lamp. Pure KBr was used as a blank control. The mixture was compressed into translucent particles using a hydraulic compressor. Soil pellets were characterized using a BioRAD FTS 135 spectrometer (Bio-Rad Corp., Hercules, CA, USA). The FTIR spectrum was recorded using a 4000–400 $cm^{-1}$ region, with 16 scans and a wavenumber resolution of 4 $cm^{-1}$. To quantify the relative changes in spectra, the relative area value of each peak was divided by the sum of the relative intensities of all peaks and multiplied by 100 using PeakFit v4.12 (Inpixon, Palo Alto, CA, USA) curve fitting software [26]. Infrared spectra were smoothed, and the baseline was corrected. The significant infrared absorption bands and organic functional groups recorded in this study are listed in Table 1.

**Table 1.** Organic functional groups absorption peaks and assignments.

| Peak | Organic Carbon Assignment | References |
| --- | --- | --- |
| 1034 | C-O stretching of polysaccharide in SOM | [10] |
| 1081 | C-O stretch in polysaccharides and fresh plant residues in DOM | [27] |
| 1428 | Symmetric stretching vibrations of the carboxylate group | [28] |
| 1625 | Aromatic C=C stretching of carboxylic acids and amides | [29] |
| 1797 | C-O stretching of carbonate | [30] |
| 2930 | Aliphatic C-H stretching of methyl and methylene groups | [31] |
| 3424 | O-H stretching for intramolecular hydrogen bonds | [32] |

### 2.5. X-ray Diffraction Analysis

The crystal structures of the rhizosphere and non-rhizosphere soils were examined using X-ray diffraction (XRD; PANalytical X'Pert PRO, Malvern Panalytical, Almelo, The Netherlands). The equipment employed Cu-Kα radiation (λ = 0.1541 nm) at an accelerating voltage of 40 kV and a beam current of 40 mA, and XRD spectra for the 2θ range of 5–80° were obtained at increments of 0.02°.

### 2.6. Statistical Analyses

SPSS Statistics 19 software (IBM Corp., Armonk, NY, USA) and Microsoft Excel 2019 (Microsoft Corp., Redmond, WA, USA) were used to analyze the properties of the rhizosphere and non-rhizosphere soils, heavy metal concentrations, microbial diversity, and relative area value of organic functional groups. To verify whether the dependent variable follows a normal distribution, the Shapiro–Wilk test was used for the raw data, and Levene's test was used to test homogeneity of variance. A one-way analysis of variance was used to determine significant differences ($p < 0.05$), and the least significant difference test was used to analyze multiple comparisons. The microbial diversity of the soil samples was calculated using QIIME software (http://qiime.org). Linear discriminant analysis effect size (LEfSe) was used to recognize the good biomarkers among groups [33], as well as to evaluate the impact of biomarkers on significantly different groups based on linear discriminant analysis (LDA) scores. Pearson's correlation analysis was used to investigate the correlation between soil factors and genus-level dominant microbial components. The relationships among soil heavy metal contents, chemical properties, organic functional groups, and genus-level microbial characteristics were examined using the redundancy analysis (RDA) in R (v4.1.0) package vegan [34].

## 3. Results and Discussion

### 3.1. The Contents of SOM and DOM

The SOM is a major index of soil fertility and a key component of soil nutrients that promotes plant growth. The SOM and DOM contents in the rhizosphere and non-rhizosphere soils of Moso bamboo forests under different cultivation patterns in heavy

metal-contaminated soil are shown in Figure 1. No difference in the SOM contents of rhizosphere and non-rhizosphere soils in bamboo monocropping was observed. In contrast, in the intercropping system, the SOM of NRI was higher than that of RI ($p < 0.01$), which is closely related to the removal of heavy metals. The ability of Moso bamboo to absorb heavy metals was enhanced in the intercropping system, and the contents of Cu, Zn, and Cd in NRI were significantly higher than those in RI (Table S1). When planting occurs in areas contaminated by heavy metals, their concentrations increase in the root zone, and the plant system absorbs them along with nutrients and accumulate in the plant. If the concentrations of heavy metals in the soil are high, their transport may compete with those of the nutrients. Therefore, heavy metals must be fixed by metal chelators in the soil solution to avoid their competition with nutrient transport [35]. Although DOM accounts for a small proportion of the total organic matter (NRM: 1.7%; NRI: 1.9%; RM: 2.2%; RI: 2.5%), as the most active component of SOM, it can chelate heavy metals, thus changing their activities and influencing their migration and transformation. This study showed that intercropping increased the content of DOM compared to monocropping, which is consistent with the findings of other studies [36,37]. The DOM acts as a substrate for microorganisms to synthesize and promote organic matter with stability [38] and can significantly improve the extractability and availability of metals, thereby enhancing their absorption by bamboo.

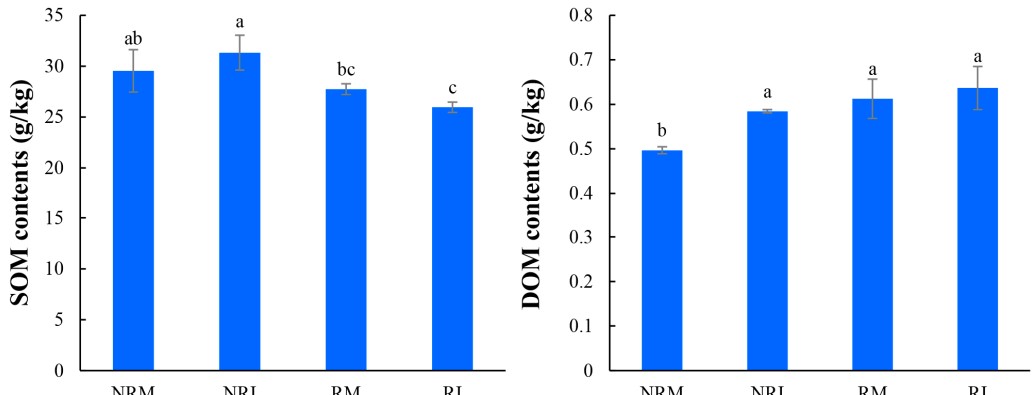

**Figure 1.** The soil and dissolved organic matter contents in rhizosphere and non-rhizosphere soils of bamboo forests. The error bar represents the standard deviation based on the triplicate tests. Different lowercase letters indicate significant differences among different types of bamboo forest soils ($p < 0.05$). NRM: non-rhizosphere soils of bamboo in the monocropping system; NRI: non-rhizosphere soils of bamboo in the intercropping system; RM: rhizosphere soils of bamboo in the monocropping system; RI: rhizosphere soils of bamboo in intercropping system.

### 3.2. Chemical Components of SOM and Soil Structure

The characteristic peaks of the soils are illustrated in Figure 2A, and the calculated intensities of the peaks are presented in Table 2. In order to quantify the relative change of FTIR spectra of bamboo forest soil organic matter, the value of the relative intensity of each peak has been widely used [10,39]. The FTIR spectra present a peak at 3424 cm$^{-1}$, indicating the occurrence of O-H stretching of intramolecular hydrogen bonds between cellulose chains [32]. The four soil types did not show any difference at 3424 cm$^{-1}$, indicating that the phytoremediation process did not increase the hydrophilicity of the metals. Hydrophilic molecules in the SOM can form strong complexes with active metals [16]. Results of this study show that the SOM also promotes the transport and uptake of metals in plants, which was confirmed by the significant differences at 1081 cm$^{-1}$, which were caused by the C-O stretching in polysaccharides and fresh plant residues in DOM [27]. RI had the highest relative proportion of polysaccharides C-O compared to those of other groups ($p < 0.01$), followed by M ($p < 0.01$). The C-O in the rhizosphere organic matter was mainly composed of root exudates, which are rich in fresh protein and lignin. The formation of such C-O in the non-rhizosphere soil increases with soil age due to increased degradation of organic

material [40]. The peak at 1034 cm$^{-1}$ was due to the C-O stretching of polysaccharide-dominated SOM [10] and did not differ among the four soil types.

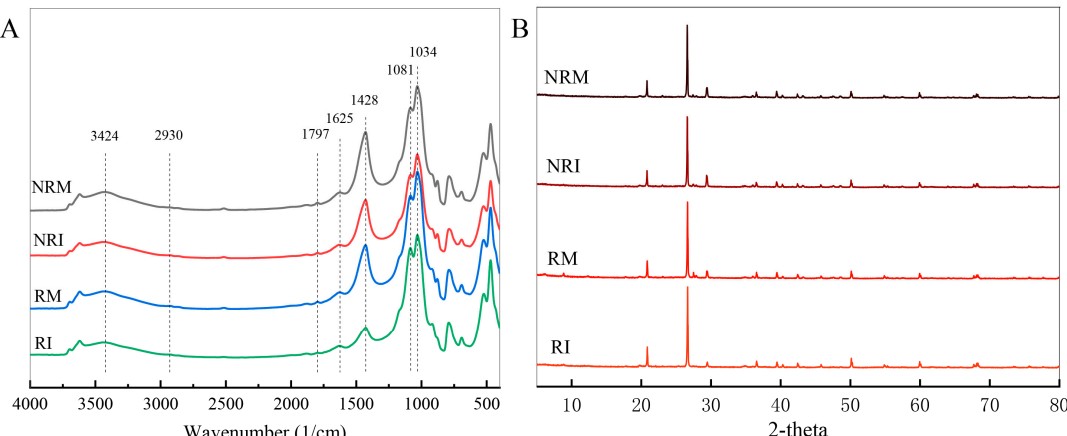

**Figure 2.** (**A**) Fourier transform infrared spectra and (**B**) X-ray diffraction spectrum of rhizosphere and non-rhizosphere soils in bamboo forest with various cultivation patterns. NRM: non-rhizosphere soils of bamboo in the monocropping system; NRI: non-rhizosphere soils of bamboo in the intercropping system; RM: rhizosphere soils of bamboo in the monocropping system; RI: rhizosphere soils of bamboo in intercropping system.

**Table 2.** The intensity of area peaks determined by FTIR spectroscopy in different soils.

| Soil Types | Peak 1034 | Peak 1081 | Peak 1428 | Peak 1625 | Peak 1797 | Peak 2930 | Peak 3424 |
|---|---|---|---|---|---|---|---|
| NRM | 26.1 ± 1.4 [a] | 21.4 ± 1.2 [c] | 25.1 ± 0.2 [a] | 6.3 ± 0.1 [a] | 2.3 ± 0.1 [b] | 3.4 ± 0.1 [b] | 5.8 ± 0.2 [a] |
| NRI | 26.2 ± 0.6 [a] | 21.0 ± 0.4 [c] | 23.9 ± 0.4 [a] | 6.5 ± 0.0 [a] | 2.5 ± 0.0 [b] | 3.7 ± 0.2 [a b] | 6.5 ± 1.1 [a] |
| RM | 26.2 ± 0.7 [a] | 25.1 ± 0.7 [b] | 20.8 ± 0.2 [b] | 6.0 ± 0.1 [a] | 2.4 ± 0.0 [b] | 3.4 ± 0.2 [b] | 6.1 ± 1.2 [a] |
| RI | 26.4 ± 1.4 [a] | 33.9 ± 2.2 [a] | 11.8 ± 1.7 [c] | 3.5 ± 2.2 [b] | 3.9 ± 1.2 [a] | 4.0 ± 0.1 [a] | 6.3 ± 0.5 [a] |

The error bar represents the standard deviation based on the triplicate tests. Different lowercase letters indicate significant differences among different types of bamboo forest soils ($p < 0.05$). NRM: non-rhizosphere soils of bamboo in the monocropping system; NRI: non-rhizosphere soils of bamboo in the intercropping system; RM: rhizosphere soils of bamboo in the monocropping system; RI: rhizosphere soils of bamboo in intercropping system.

The peak at 1797 cm$^{-1}$ is attributed to the C-O stretching of carbonates [30]. The peak at 2930 cm$^{-1}$ is caused by the aliphatic C-H stretching of methyl groups and methylene [31]. RI had a high relative proportion of carbonate C-O, aliphatic methyl, and methylene ($p < 0.05$), whereas the other treatments showed no difference. This suggests that the SOM in RI contained more saturated and oxidized compounds [16]. The transmittance at 1428 cm$^{-1}$ corresponded to asymmetric and symmetric stretching caused by the presence of carboxylate ions, which increased proportionally with the increase in heavy metal concentration in the soil [28]. The 1625 cm$^{-1}$ peak is due to the C=C stretching in aromatics [29]. NRM and NRI had a high proportion of symmetric carboxylates and aromatic C=C ($p < 0.05$), followed by M, while I had the lowest proportion ($p < 0.05$). This may be attributed to the higher degradation of SOM in the non-rhizosphere soil, which increases the stability and saturation of molecules [16,40]. The results of this study showed that the proportion of soil C of Moso bamboo increased in the intercropping system compared to that of the monocropping system (Table 2). Therefore, the Moso bamboo forest ecosystem acts as a C sink and plays an important role in the phytoremediation of heavy metals, and this ability was particularly effective in the intercropping system. The mechanism of promoting C accumulation by the bamboo soil is mainly related to C stability.

The XRD spectra of all bulk components indicated that silica occupied the majority of both rhizosphere and non-rhizosphere soils, although other crystalline components were also present (Figure 2B). While the four curves exhibited similar peaks, there were some variations in the details. We observed that the characteristic peaks of NRM, NRI, and RM

were located at 2θ = 27°, indicating the presence of structured C, which is consistent with the findings of Senneca, et al. [41] for metal-contaminated soil. The peaks of NRM and NRI at 2θ = 29°, with noticeable broadened signals and expanded regions, highlight key discrepancies that occur when Cu and Zn ions change [42]. The precipitation of Cd ions at this peak was also observed in the Cd-sorbed material [43]. Changes in intramolecular and intermolecular bonds occurred in the XRD spectral bands under the influence of different soil concentrations of Cd, Cu, and Zn.

### 3.3. Soil Bacterial Community

From the studied soil samples, after the initial quality control, we obtained a total of 325,259 high-quality sequences with a mean of 27,105 ± 855 sequences per sample, distributed among 21,591 operational taxonomic units. The bacterial diversity indices of the non-rhizosphere soil were higher than those of the rhizosphere soil, especially the phylogenetic diversity whole tree and Chao1 indices (Figure 3). The bacterial community in the bamboo rhizosphere changed noticeably under different planting systems, which is consistent with research that shows that bacterial communities in the rhizosphere and non-rhizosphere soils differ in structure and diversity due to the influence of environmental factors and cultivation patterns [21].

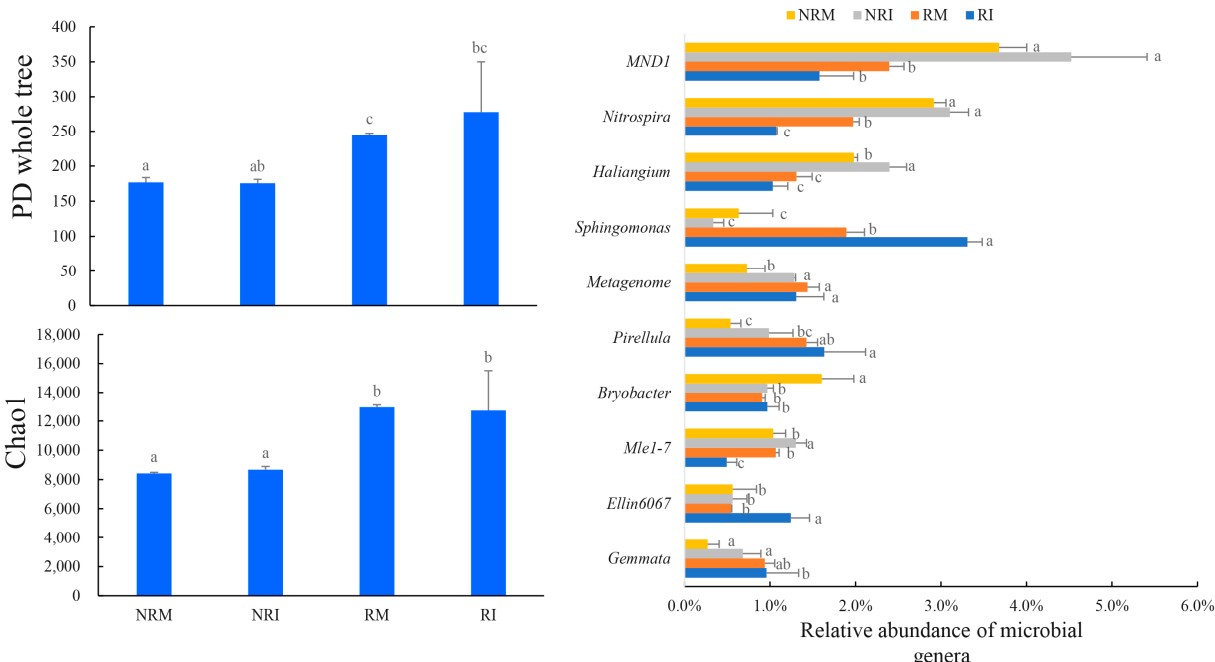

**Figure 3.** Microorganisms in rhizosphere and non-rhizosphere soils of bamboo forests. The error bar represents the standard deviation based on the triplicate tests. Different lowercase letters indicate significant differences among different types of bamboo forest soils ($p < 0.05$). NRM: non-rhizosphere soils of bamboo in the monocropping system; NRI: non-rhizosphere soils of bamboo in the intercropping system; RM: rhizosphere soils of bamboo in the monocropping system; RI: rhizosphere soils of bamboo in intercropping system.

Proteobacteria (25.4%–37.7%), Acidobacteria (16.4%–27.9%), Planctomycetes (6.9%–17.3%), Gemmatimonadetes (3.6%–15.4%), and Actinobacteria (3.2%–9.2%) were the top five most abundant phyla, accounting for approximately 78.7% of the 16S rRNA gene abundance. Across all soil samples, the relative abundances of Verrucomicrobia, Bacteroidetes, Chloroflexi, Nitrospirae, Rokubacteria, Latescibacteria, and Patescibacteria were >1%. These phyla, particularly Proteobacteria, which often have a higher relative abundance in stressful environments, are the most active bacteria in the rhizosphere soil [7,44]. This is probably a result of the niche occupied by synergistic effects, such as co-metabolism or synthesis, in

response to environmental, biological, chemical, or physical factors [45]. The results suggest that Actinobacteria, Gemmatimonadetes, and Nitrospirae are well-suited for survival in nutrient-rich non-rhizosphere soils, and these can also withstand heavy metal stress in soils, which is consistent with the results of previous studies [46,47].

The top 10 bacterial genera with an average relative abundance > 0.7% are shown in Figure 3. The bacteria affecting soil nutrient cycling, *MND1*, *Nitrospira*, *Haliangium*, and *Bryobacter* [48,49], were more abundant in the non-rhizosphere soil than in the rhizosphere soil ($p < 0.01$). Among them, *Bryobacter* had the highest relative abundance in the NRM treatment ($p < 0.01$). *Sphingomonas* is a novel and abundant microorganism that can survive in environments with limited nutrients and exhibits characteristics of denitrification and non-symbiotic N fixation [50]. The large root system of Moso bamboo has dense and fine roots that extract heavy metals and obtain nutrients from the rhizosphere soil, resulting in lower nutrients in the rhizosphere soil than those in the non-rhizosphere soil. *Sphingomonas* and *Pirellula* abundance were found to be lower in the rhizosphere soil than in the non-rhizosphere soil ($p < 0.05$), indicating that these two genera can survive in soils with low nutrient levels. *Sphingomonas* and *Ellin6067* (an ammonia-oxidizing bacterium that converts ammonia to nitrite [51]) were the highest in RI ($p < 0.01$). Using cover crops in intercropping increased the relative abundance of potentially plant-beneficial bacteria [49].

LEfSe can be used to determine the genetic or functional characteristics that best explain the differences and quantify their effects between groups under different treatments [52]. Among the four soil types tested using LefSe, intercropping had more species than monocropping, with LDA scores of >4; that is, NRI (11) > NRM (10), RI (11) > RM (6). In addition, we found that intercropping bamboo under heavy metal stress produced the most biomarkers at the genus level. *MND1* and *Nitrospira* in the NRI treatment and *Sphingomonas* in the RI treatment most effectively explained the variations between bamboo in rhizosphere and non-rhizosphere soils with intercropping and monocropping. These bacteria may play a role in the N cycle [53]: *MND1* and *Nitrospira* function as nitrifiers, while the N-fixing bacteria *Sphingomonas* can enhance the production of $NO^{3-}$-N. In addition, they have also been proven to have strong tolerance to heavy metals [54–56]: *Nitrospira* exhibits Cd tolerance through the overproduction of electron-transporting cytochrome c-like proteins, while *Sphingomonas*, a plant growth-promoting rhizobacterium, enhances plant productivity by hormone production. This suggests that after *S. plumbizincicola* was planted, bacteria involved in heavy metal remediation and nutrient cycling were enriched in the bamboo forest.

*3.4. Relationships among Soil Heavy Metals, Organic Components, and Microorganisms*

Correlation analysis and RDA have been widely used to test the relationship between environmental factors and microorganisms [57]. The relative abundance of bacterial genera *Sphingomonas* and *Gemmata* were significantly positively correlated with heavy metals (Figure 4A). However, the abundance of *MND1*, *Nitrospira*, *Haliangium*, *Bryobacter*, and *Mle1-7* was negatively correlated with heavy metals. The abundance of *Nitrospira* and *Bryobacter* was positively correlated with polysaccharides C-O. In contrast, the abundance of *Sphingomonas*, *Metagenome*, *Pirellula*, and *Gemmata* were negatively correlated with polysaccharides C-O. These genera were inversely related to symmetric carboxylate. *Gemmata* has been reported to remove $NH_4$-N and heavy metals from pollutants [58], and in this study, it was related to the organic functional groups except for aliphatic methyl and methylene. Organic functional groups are strongly correlated with *Bryobacter*, which can grow under anoxic conditions using various fatty acids, sugars, and polysaccharides that can reduce nitrate [59], thereby playing an important role in the process of restoration of vegetation [60]. The dominant microbial genera in the rhizosphere and non-rhizosphere soils were closely correlated with the organic functional groups of SOM and were involved in soil nutrient cycling across all vegetation restoration processes.

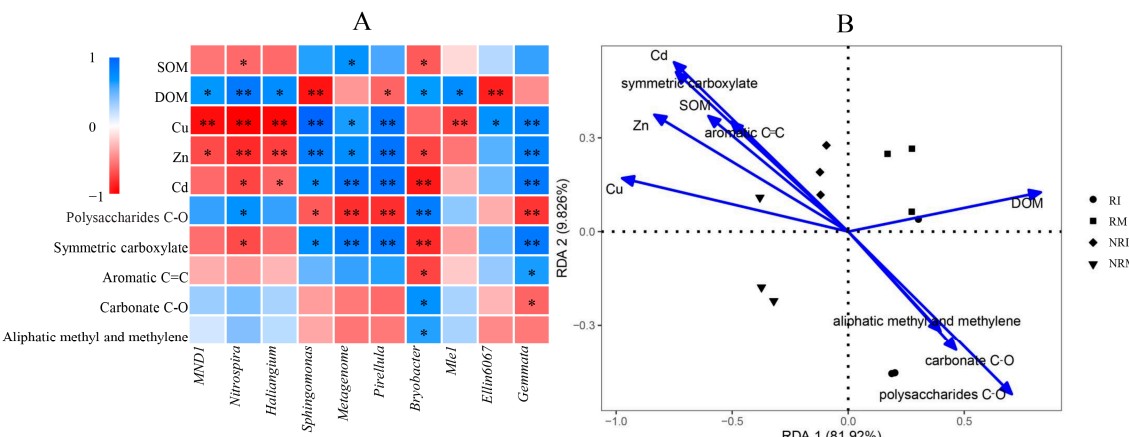

**Figure 4.** Relationships among soil heavy metal contents, chemical properties, organic functional groups, and microbial characteristics. (**A**) correlation analysis; (**B**) redundancy analysis. SOM: Soil organic matter; DOM: dissolved organic matter. *: $p < 0.05$; **: $p < 0.01$.

The RDA method was employed to study the relationship between the main soil factors that affect the bacterial community structure at the OTU level. The results of the RDA showed a definite separation between the four soil types (Figure 4B). The first and second axes explained 81.92% and 9.826% of the total variation, respectively. The ordination axis revealed significant correlations between soil characteristics and total bacterial community structure (Table 3), particularly Cu, Zn, Cd, SOM, DOM, polysaccharides C-O, and symmetric carboxylate. Previous studies have shown that heavy metals can alter the bacterial community structure [7,19]. Heavy metals cause single and double-strand breaks in organisms that modify their bases, causing changes in the bacterial community [57]. Functional groups in the SOM react with heavy metal ions to form stable organic chelates containing heavy metals. In turn, this increases the bioavailability and mobility of heavy metals in soil and facilitates their extraction.

**Table 3.** Correlations between soil characteristics and microbial community with the ordination axis.

| Index | RDA1 | RDA2 | $r^2$ | $p$ | |
|---|---|---|---|---|---|
| SOM | −0.852 | 0.524 | 0.500 | 0.046 | * |
| DOM | 0.989 | 0.149 | 0.704 | 0.005 | ** |
| Cu | −0.985 | 0.174 | 0.977 | 0.001 | *** |
| Zn | −0.912 | 0.409 | 0.838 | 0.002 | ** |
| Cd | −0.810 | 0.586 | 0.858 | 0.001 | *** |
| Polysaccharides C-O | 0.804 | −0.595 | 0.768 | 0.003 | ** |
| Symmetric carboxylate | −0.824 | 0.567 | 0.811 | 0.002 | ** |
| Aromatic C=C | −0.820 | 0.572 | 0.379 | 0.072 | |
| Carbonate C-O | 0.774 | −0.633 | 0.360 | 0.086 | |
| Aliphatic methyl and methylene | 0.779 | −0.627 | 0.261 | 0.239 | |

SOM: Soil organic matter; DOM: dissolved organic matter. *: $p < 0.05$; **: $p < 0.01$; ***: $p = 0.001$.

Previous studies have shown that the high content of polysaccharides is the ideal structural characteristic of water-extractable organic matter to promote the formation and stability of macroaggregates [38]. Carboxylates can mobilize soil heavy metal elements and stabilize SOM by increasing the exposure of inaccessible C in the soil through a real priming effect [61]. Our study clarified that SOM and its refined organic functional groups, especially polysaccharides C-O and symmetric carboxylate, significantly impacted the bacterial population structure and SOM stabilization during heavy metal remediation in bamboo forests. The SOM provides organic chemicals as chelators, and the interaction of soil microbial communities is the main mechanism that promotes heavy metal remediation.

*3.5. New Shoot Growth*

According to soil pollution in agricultural land in China [62], it was found that the soil contained significantly elevated levels of Zn and Cd, surpassing the risk screening values (pH > 7.5; Zn: 300, Cd: 0.8 mg/kg). Implementing a phytoremediation model involving the continuous harvesting of bamboo and the cultivation of new bamboo plants has proven to be an effective method for removing heavy metals from the soil. Intercropping bamboo can contribute to the accumulation of SOM, which increases the capacity of the plants for the absorption, transport, and storage of heavy metals. Soil organic functional groups coordinate and complex heavy metal ions to form organic chelates that are absorbed and transported by plant roots and accumulate in stems and leaves, thereby reducing the biotoxicity of heavy metals to plants and improving plant biomass [63]. Research on microalgae has revealed that heavy metals can enter plant cells either through active transportation or endocytosis facilitated by plant chelating proteins, thereby mitigating the toxicity of heavy metals. However, when an excess of heavy metals accumulates, heavy metal ions bind to thiol groups within protein cysteine, disrupting the protein structure or displacing essential elements [64]. The emergence of new shoots indicates healthy growth and robust stress resistance in plants. During the field investigation, the density of bamboo shoots in the intercropped bamboo forest was $2.6 \pm 0.3$ per m$^2$, while that in the pure bamboo forest was $0.8 \pm 0.2$ per m$^2$, indicating that with a decrease in heavy metal concentration in intercropped soil, the environment became more suitable for sprouting of bamboo shoots. This contributes to the increase in productivity of the bamboo forest, as the Moso bamboo–*S. plumbizincicola* intercropping stands have higher productivity than pure Moso bamboo forests.

Intercropping of maize and pea can increase crop yields, and an analysis of soil respiration rates shows that intercropping systems have high C sequestration potential (net primary production/C emission = 2.01) and economic benefits [65]. This is also applicable in the bamboo–*S. plumbizincicola* intercropping system. Although more bamboo shoots appeared in the intercropping stand, Figure 5 illustrates that the heavy metals in bamboo shoots in intercropping stands were significantly higher than those in monocropping stands ($p < 0.01$). According to China's national standard for food safety [66], the permissible limit of Cd in bamboo shoots is 0.05 mg/kg. Evidently, our study's findings indicate that the heavy metal content in bamboo shoots surpasses the safe consumption standard. Consequently, the bamboo forest intended for heavy metal restoration is reclassified as a timber forest. The biomass resulting from phytoremediation, rich in heavy metals, can be utilized to obtain biofuels and biochar and for phyto-mining to recover specific valuable metals [67]. Intercropping of bamboo is an effective way to improve plant tolerance to heavy metal ion toxicity and change the rhizosphere environment, thus facilitating the extraction and transport of heavy metals by roots and their migration to new shoots and other aboveground parts of plants. High levels of C, N, and phosphorus (P) have a great support effect on plant growth [68], and the enhanced absorption of CNP is directly related to increased photosynthesis. Notably, the phenomenon of low P and rich N is commonly observed in Moso bamboo forest cultivation. Subsequent studies should prioritize investigating the CNP ratio in phytoremediation soil, as it benefits bamboo growth and development under low P.

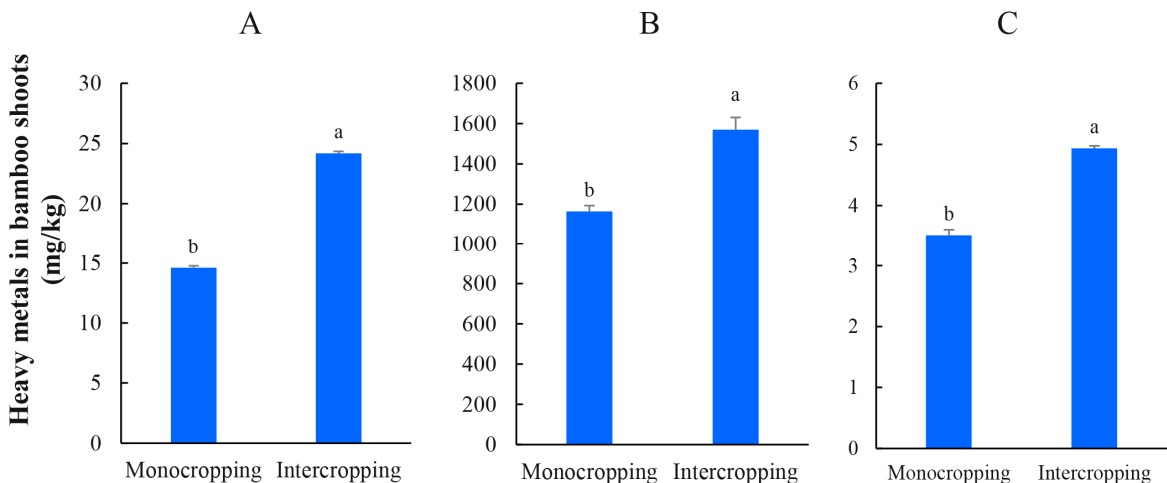

**Figure 5.** Heavy metal content of new Moso bamboo shoots with different cultivation patterns. (**A**) Cu; (**B**) Zn; (**C**) Cd. The error bar represents the standard deviation based on triplicate tests. Different lowercase letters indicate significant differences in the heavy metal contents of bamboo shoots between different cultivation patterns ($p < 0.05$).

## 4. Conclusions

This study investigated the qualitative and quantitative differences in soil organic matter in Moso bamboo forests under different cultivation patterns to better characterize and comprehend the behavior of heavy metals during phytoremediation. The contents of SOM and DOM in rhizosphere and non-rhizosphere soils were significantly different between monocropping and intercropping systems. Intercropping increased DOM content in the soil of bamboo forests, which might enhance the mobility of heavy metals and increase their availability for plant absorption. A close relationship between organic functional groups and genus-level dominant microorganisms, such as *Sphingomonas*, *Gemmata*, and *Bryobacter*, was observed. The bacterial community structure was significantly correlated with polysaccharide C-O and symmetric carboxylates. These functional groups are related to the soil C cycle and play important roles in increasing the efficiency of heavy metal remediation in intercropping systems. The decrease in heavy metal concentrations in the intercropping soil was suitable for bamboo shoot sprouting and forest production. Our findings demonstrate that improving the storage of organic C and organic matter in the soil of Moso bamboo forests during the phytoremediation of heavy metal contamination offers an applicable, innovative, feasible, and sustainable restoration method. Future research is needed to explore the synergistic sustainable development of C sequestration, soil pollution remediation, and soil health maintenance.

**Supplementary Materials:** The following supporting information can be downloaded at https://www.mdpi.com/article/10.3390/f14091895/s1, Table S1: Basic chemical properties of non-rhizosphere and rhizosphere soil in Moso bamboo forest.

**Author Contributions:** Conceptualization and experimental design, F.B. and Z.Z.; Methodology, X.Z.; Investigation, X.Z., Z.H. and Q.L.; Writing—original draft preparation, F.B.; Writing—review and editing, F.B. and Z.Z.; Funding acquisition, F.B. All authors have read and agreed to the published version of the manuscript.

**Funding:** This research was supported by the Zhejiang Provincial Natural Science Foundation of China under Grant No. LTGS23C160002, the National Natural Science Foundation of China under Grant No. 32201538, and the Talent Development Program of China National Bamboo Research Center under Grant No. ZXYC202202.

**Acknowledgments:** The authors wish to thank the anonymous referees for their valuable comments.

**Conflicts of Interest:** The authors declare no conflict of interest.

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
