# Peer review of "Enhancement of Phytoremediation of Heavy Metal Pollution Using an Intercropping System in Moso Bamboo Forests: Characteristics of Soil Organic Matter and Bacterial Communities"

_forests, doi:10.3390/f14091895_

Round 1

Reviewer 1 Report

Interesting research!

Abstract can be improved. State the problem statement and the research gap clearly. Significant findings should be emphasized. Recommendation for future studies must be included. Arrange keywords in alphabetical order.

State the model and manufacturer of major equipment.

Is the Maso bamboo safe for use/consumption after absorbing the heavy metals? Any experiments conducted to verify this concern?

Which N:P:C ratio supported the best phytoremediation of the heavy metals and please explain. Reference:

(a) Yaacob, N. S., Ahmad, M. F., Kawasaki, N., Maniyam, M. N., Abdullah, H., Hashim, E. F., ... & Kuwahara, V. S. (2021). Kinetics Growth and Recovery of Valuable Nutrients from Selangor Peat Swamp and Pristine Forest Soils Using Different Extraction Methods as Potential Microalgae Growth Enhancers. Molecules26(3), 653.

Did the authors tested the ability of the abundant microbes for heavy metal bioremediation?

Mostly recent references are used. 

Minor editing is required. 

Author Response

Dear Ms. Janina Yin and Reviewers,

Thank you for your letter and for the Reviewers’ comments concerning our manuscript (Number: forests-2608494). Those comments are all valuable and very helpful for revising and improving our paper. Revised portions are marked in red in the paper. Point by point responses to the reviewers’ comments are as flowing:

Responds to the reviewer’s comments

Responses to Reviewer 1:

  1. Abstract can be improved. State the problem statement and the research gap clearly. Significant findings should be emphasized. Recommendation for future studies must be included. Arrange keywords in alphabetical order.

Response: We have rewritten the abstract as follows:

Abstract: Heavy metal pollution in soil is a major global issue, and one effective method for addressing it is phytoremediation through bamboo planting. Nevertheless, there is a notable gap in our knowledge as no studies have explored the characteristics of soil organic matter (SOM) and the bacterial communities in bamboo forests during the remediation process. To bridge this knowledge gap, we conducted research to investigate the impact of different bamboo planting patterns on the SOM characteristics and microbial communities in soils contaminated with heavy metals. The contents of SOM and dissolved organic matter (DOM) in rhizosphere and non-rhizosphere soils differed significantly between monocropping and intercropping systems, with DOM accounting for only 1.7–2.5% of SOM. Fourier transform infrared spectra showed that the contents of SOM polysaccharides C-O, carbonate C-O, aliphatic methyl, and methylene increased, while the aromatic C=C abundance decreased in the intercropping rhizo-sphere soil. The differences between bamboo cultivation patterns in the rhizosphere and non-rhizosphere soils were elucidated using the biomarkers, including MND1 and Nitrospira (non-rhizosphere), and Sphingomonas (rhizosphere). Heavy metals, DOM, SOM, and refined organic functional groups, especially C-O in polysaccharides and symmetric carboxylate, were the determining factors of soil bacterial communities. Compared to monocropping, intercropping increased the accumulation of Zn and Cd in bamboo shoots by 35% and 40%, respectively, and hence, intercropping soil, with a low toxicity, was suitable for bamboo shoot sprouting. Intercropping can alter the characteristics of SOM and bacterial communities and plays a vital role in phytoremediation and shoot growth in bamboo forests. Future studies on soil carbon dynamics and nutrient status during heavy metal remediation will improve our knowledge of soil transformation and its impact on soil ecosystem health and productivity.

Keywords: Bacterial community; bamboo forest; heavy metal pollution; intercropping; phy-toremediation; soil organic matter.

  1. State the model and manufacturer of major equipment.

Response: We have supplemented the model and manufacturer of major equipment: “After the soil was digested using HNO3–HCl–HClO4, the heavy metal content in the soil was determined using atomic absorption spectrophotometry (AAnalyst800, Perkin Elmer, USA)”, “Bamboo shoots were dried and ashed, and their heavy metal contents were determined using atomic absorption spectrophotometry (AAnalyst800, Perkin Elmer, USA)”, “The PCR products were sequenced on the Illumina MiSeq platform (Illumina Inc., San Diego, CA, USA) by G-BIO Technology Co., Ltd. (Hangzhou, China).”

  1. Is the Maso bamboo safe for use/consumption after absorbing the heavy metals? Any experiments conducted to verify this concern?

Response: We have added the information of the hygiene requirements for heavy metals in plants for China and the utilization of phytoremediation biomass rich in heavy metals: “According to China's national standard for food safety [63], the permissible limit of Cd in bamboo shoots is 0.05 mg/kg. Evidently, our study's findings indicate that the heavy metal content in bamboo shoots surpasses the safe consumption standard. Consequently, the bamboo forest intended for heavy metal restoration is reclassified as a timber forest. The biomass resulting from phytoremediation, rich in heavy metals, can be utilized to obtain biofuels, biochar, and for phyto-mining to recover specific valuable metals [64].”.

Newly added references

GB2762-2022. China National Food Safety Standard Maximum Levels of Contaminants in Food. 2022.

Quarshie, S.D.-G.; Xiao, X.; Zhang, L. Enhanced phytoremediation of soil heavy metal pollution and commercial utilization of harvested plant biomass: a review. Water, Air, Soil Pollut. 2021, 232, 1-28, doi:10.1007/s11270-021-05430-7.

  1. Which N:P:C ratio supported the best phytoremediation of the heavy metals and please explain. Reference:

(a) Yaacob, N. S., Ahmad, M. F., Kawasaki, N., Maniyam, M. N., Abdullah, H., Hashim, E. F., ... & Kuwahara, V. S. (2021). Kinetics Growth and Recovery of Valuable Nutrients from Selangor Peat Swamp and Pristine Forest Soils Using Different Extraction Methods as Potential Microalgae Growth Enhancers. Molecules, 26(3), 653.

Response: We have added the discussion about mechanisms of plant resistance to metal toxicity: “Soil organic functional groups coordinate and complex heavy metal ions to form organic chelates that are absorbed and transported by plant roots and accumulate in stems and leaves, thereby reducing the biotoxicity of heavy metals to plants and improving plant biomass [60]. Research on microalgae has revealed that heavy metals can enter plant cells either through active transportation or endocytosis, facilitated by plant chelating proteins, thus mitigating the toxicity of heavy metals. However, when an excess of heavy metals accumulates, heavy metal ions bind to thiol groups within protein cysteine, resulting in the disruption of protein structure or the displacement of essential elements [61].”

And added the discussion about soil CNP: “High levels of C, N, and phosphorus (P) have a great support effect on plant growth [65], and the enhanced absorption of CNP is directly related to increased photosynthesis. Notably, the phenomenon of low P and rich N is commonly observed in Moso bamboo forest cultivation. Subsequent studies should prioritize investigating the CNP ratio in phytoremediation soil, as it benefits bamboo growth and development under low P.”

Newly added references

Yaacob, N.S.; Ahmad, M.F.; Sivam, A.; Hashim, E.F.; Maniyam, M.N.; Sjahrir, F.; Dzulkafli, N.F.; Wan Mohd Zamri, W.M.I.; Komatsu, K.; Kuwahara, V.S.; et al. The Effectiveness of Soil Extracts from Selangor Peat Swamp and Pristine Forest Soils on the Growth of Green Microalgae sp. Forests 2022, 13, 79, doi:10.3390/f13010079.

Yaacob, N.S.; Ahmad, M.F.; Kawasaki, N.; Maniyam, M.N.; Abdullah, H.; Hashim, E.F.; Sjahrir, F.; Wan Mohd Zamri, W.M.I.; Komatsu, K.; Kuwahara, V.S. Kinetics Growth and Recovery of Valuable Nutrients from Selangor Peat Swamp and Pristine Forest Soils Using Different Extraction Methods as Potential Microalgae Growth Enhancers. Molecules 2021, 26, 653, doi:10.3390/molecules26030653.

  1. Did the authors tested the ability of the abundant microbes for heavy metal bioremediation?

Mostly recent references are used.

Response: Thank you for the advice. At present, we have not tested the ability of microbes on heavy metals bioremediation. Instead, we have searched relevant references to support the effect of detected dominant microbes on heavy metal remediation. And recent references are used: “MND1 and Nitrospira in the NRI treatment and Sphingomonas in the RI treatment most effectively explained the variations between bamboo in rhizosphere and non-rhizosphere soils with intercropping and monocropping. These bacteria may play a role in the nitrogen cycle [50]: MND1 and Nitrospira function as nitrifiers, while the N-fixing bacteria Sphingomonas can enhance the production of NO3− -N. In addition, they have also been proven to have strong tolerance to heavy metals [51-53]: Nitrospira exhibits Cd tolerance through the overproduction of electron-transporting cytochrome c-like proteins, while Sphingomonas, a plant growth -promoting rhizobacterium, enhances plant productivity by hormone production.”

Newly added references

Wang, Q.; Feng, X.; Liu, Y.; Cui, W.; Sun, Y.; Zhang, S.; Wang, F. Effects of microplastics and carbon nanotubes on soil geochemical properties and bacterial communities. J. Hazard. Mater. 2022, 433, 128826, doi:10.1016/j.jhazmat.2022.128826.

Nagarajan, V.; Tsai, H.-C.; Chen, J.-S.; Hussain, B.; Koner, S.; Hseu, Z.-Y.; Hsu, B.-M. Comparison of bacterial communities and their functional profiling using 16S rRNA gene sequencing between the inherent serpentine-associated sites, hyper-accumulator, downgradient agricultural farmlands, and distal non-serpentine soils. J. Hazard. Mater. 2022, 431, 128557, doi:10.1016/j.jhazmat.2022.128557.

Zhang, M.; Xiong, J.; Zhou, L.; Li, J.; Fan, J.; Li, X.; Zhang, T.; Yin, Z.; Yin, H.; Liu, X. Community ecological study on the reduction of soil antimony bioavailability by SRB-based remediation technologies. J. Hazard. Mater. 2023, 132256, doi:10.1016/j.jhazmat.2023.132256.

Yang, Z.-N.; Liu, Z.-S.; Wang, K.-H.; Liang, Z.-L.; Abdugheni, R.; Huang, Y.; Wang, R.-H.; Ma, H.-L.; Wang, X.-K.; Yang, M.-L. Soil microbiomes divergently respond to heavy metals and polycyclic aromatic hydrocarbons in contaminated industrial sites. Environmental Science and Ecotechnology 2022, 10, 100169, doi:10.1016/j.ese.2022.100169.

In addition to the Reviewers’ comments, we have done some modification on those sections, including abstract, materials and methods, results and discussion, that were amended by Editage Editing.

Once again, thank you very much for your comments and suggestions.

Reviewer 2 Report

some comments in the attached file

Author Response

Dear Ms. Janina Yin and Reviewers,

Thank you for your letter and for the Reviewers’ comments concerning our manuscript (Number: forests-2608494). Those comments are all valuable and very helpful for revising and improving our paper. Revised portions are marked in red in the paper. Point by point responses to the reviewers’ comments are as flowing:

Responds to the reviewer’s comments

Responses to Reviewer 2:

  1. Line 93-95. The soil at the study site was characterized as a Ferralsol according to the soil classification system of the Food and Agriculture Organization of the United Nations - Can the chemical and morphological properties of soils be explained in more detail?

Response: Thank you for the advice. We have added the chemical and morphological properties of soils in more detail: “Fuyang is situated on a flat plain with predominantly fine sand and clay surface sediments dominating the landscape. The soil at the research site falls under the classification of Ferralsol, as per the soil classification system established by the Food and Agriculture Organization of the United Nations. This Ferralsol is characterized as red soil and has developed from the parent material of siltstone [23]. The average values for the basic physicochemical properties at the study site were: pH of 7.83, bulk density of 1.66 g/cm3; water content of 210.57 g/kg; total porosity of 42.25%; organic C of 25.8 g/kg; total N of 2.41 g/kg.”

  1. Line 97-98. The soil in the study site was polluted with industrial wastewater, leading to the presence of heavy metals - Can the conditions of soil contamination be explained in more detail (time of contamination, what industrial wastewaters, characterization of waters)?

Response: We have added the conditions of soil contamination in more detail: “Notably, in proximity to our research site, a hardware electroplating factory had been discharging industrial wastewater into farmland irrigation channels. However, this dis-charge practice was terminated in 2007. The electroplating factory's wastewater contains a range of heavy metal ions such as Zn, Cu, Cd, Ni, Pb, as well as other hazardous components, including acids, alkalis, cyanides, and various toxic debris. The soil at the study site was polluted with industrial wastewater, leading to the presence of heavy metals. Preliminary soil analysis indicated that the primary contaminants were Zn, Cu, and Cd.”

  1. What are the hygiene requirements for heavy metals in soil and plants for China? It is necessary to compare the heavy metal content of soil and plants in the experiment with the hygiene requirements.

Response: We have added the information of the hygiene requirements for heavy metals in soil and plants for China: “According to soil pollution in agricultural land in China [59], , it was found that the soil contained significantly elevated levels of Zn and Cd, surpassing the risk screening values (pH > 7.5; Zn: 300, Cd: 0.8 mg/kg). Implementing a phytoremediation model involving the continuous harvesting of bamboo and the cultivation of new bamboo plants has proven to be an effective method for removing heavy metals from the soil.”

And “According to China's national standard for food safety [63], the permissible limit of Cd in bamboo shoots is 0.05 mg/kg. Evidently, our study's findings indicate that the heavy metal content in bamboo shoots surpasses the safe consumption standard. Consequently, the bamboo forest intended for heavy metal restoration is reclassified as a timber forest. The biomass resulting from phytoremediation, rich in heavy metals, can be utilized to obtain biofuels, biochar, and for phyto-mining to recover specific valuable metals [64].”

Newly added references

GB 15618-2018. Soil Environmental Quality Risk Control Standard For Soil Contamination of Agricultural Land in China. 2018.

GB 2762-2022. China National Food Safety Standard Maximum Levels of Contaminants in Food. 2022.

Quarshie, S.D.-G.; Xiao, X.; Zhang, L. Enhanced phytoremediation of soil heavy metal pollution and commercial utilization of harvested plant biomass: a review. Water, Air, Soil Pollut. 2021, 232, 1-28, doi:10.1007/s11270-021-05430-7.

  1. Poor quality of figures in the text of the article. Some notations on the figures are very difficult to read.

Response: We have improved the quality of figures in the text of the article.

In addition to the Reviewers’ comments, we have done some modification on those sections, including abstract, materials and methods, results and discussion, that were amended by Editage Editing.

Once again, thank you very much for your comments and suggestions.

Round 2

Reviewer 1 Report

The paper is now ready to be published after the amendments. Well done.

Author Response

Sep 15, 2023

Forests

Dear Editor and Reviewers,

Thank you for your letter and for the reviewers’ comments concerning our manuscript (Number: forests-2608494). These comments were really valuable for revising and improving our work. Based on these insightful comments and suggestions, we have revised the manuscript carefully. We hope that all these changes fulfill the requirements to make the manuscript acceptable for publication in Forests. The revised parts are indicated in red color marked in the newly submitted version. Our point-by-point responses to the reviewers’ comments are as follows:

Responds to the comments

Minor editorial improvements could still be made to this manuscript before acceptance.

In the abstract (L21) the word rhizosphere is used hyphenated (rhizo-sphere) whereas elsewhere it is used unhyphenated (rhizosphere).

Response: Thank you for your careful review! We have unified the use of words throughout the text.

L53 - the abbreviation "C" is defined for carbon, then L54, L55 the full word carbon is used.

Response: We have made corresponding modifications throughout the text. In addition, we have changed the abbreviation “N” for full word nitrogen.

Statistical Analysis

The is no mention of testing for normality or homoscedasticity in the statistical analysis to ensure that the prerequisites of the parametric analysis (ANOVA) used are met. Please could the authors confirm that they assessed the prerequisites of ANOVA. Please also state the tests they used and note if any transformations were required. The authors also state that they used R for redundancy analysis. Please state the version of R used and reference the package name for the analysis. Was the easyCoda package v0.34.3 used?

Response: We have added the description of testing for normality or homoscedasticity before applying ANOVA in the methods/statistical analysis section: “To verify whether the dependent variable follows a normal distribution, Shaprio-Wilk test was used for the raw data, and Levene’s test was used to test homogeneity of variance. A one-way analysis of variance was used to determine significant differences (P < 0.05), and the least significant difference test was used to analyze multiple comparisons.”

And we have stated the version of R used and reference the package name for the analysis: “The relationships among soil heavy metal contents, chemical properties, organic functional groups, and genus level microbial characteristics were examined using the redundancy analysis (RDA) in R (v4.1.0) package vegan [27].”

Newly added references

Oksanen, J.; Blanchet, F.G.; Friendly, M.; Kindt, R.; Legendre, P.; McGlinn, D.; Minchin, P.R.; O'hara, R.; Simpson, G.L.; Solymos, P. Vegan: Community Ecology Package. R package version 2.5-5 2019.

Linear discriminant analysis effect size is reported at L312, however this methodology is not described in the methods/statistical analysis section.

Response: We have described Linear discriminant analysis effect size in the methods/statistical analysis section: “Linear discriminant analysis effect size (LEfSe) was used to recognize the good biomarkers among groups [27], as well as to evaluate the impact of biomarkers on significantly different groups based on linear discriminant analysis (LDA) scores.”

Newly added references

Segata, N.; Izard, J.; Waldron, L.; Gevers, D.; Miropolsky, L.; Garrett, W.S.; Huttenhower, C. Metagenomic biomarker discovery and explanation. Genome Biol. 2011, 12, 1-18, doi:10.1186/gb-2011-12-6-r60.

Figures/Tables should be interpretable using the figure/table descriptions without referring the manuscript text. Please revise all figure/table descriptions ensuring that all acronyms/abbreviations have been defined in the figure description.

Response: We have revised all figure/table descriptions ensuring that all acronyms/abbreviations in the figure/table descriptions are defined.

Please could the authors reformat the table header in Table 2 to differentiate between treatment and wavenumber. The description could also be improved to detail if SE or SD was used as well as the use of subscript letters. I'm afraid that the text "intensity of area peaks" doesn't make sense to me, particularly as the units are a percentage. Please clarify by rewording the description. n.b. treatments are defined here in a footnote, but not in figure 1.

Response: We have reformatted the table header in Table 2 to differentiate between treatment and wavenumber. The description has been improved to detail. The text "intensity of area peaks" is meaningful, and we have added references to support the idea: “In order to quantify the relative change of FTIR spectra of bamboo forest soil organic matter, the value of relative intensity of each peak has been widely used [10,33].” The treatments have been defined in a footnote of figure 1.

Newly added references

Yang, C.; Ni, H.; Zhong, Z.; Zhang, X.; Bian, F. Changes in soil carbon pools and components induced by replacing secondary evergreen broadleaf forest with Moso bamboo plantations in subtropical China. Catena 2019, 180, 309-319, doi:10.1016/j.catena.2019.02.024.

Yang, C.; Zhong, Z.; Zhang, X.; Bian, F.; Du, X. Responses of soil organic carbon sequestration potential and bacterial community structure in moso bamboo plantations to different management strategies in subtropical China. Forests 2018, 9, 657, doi:10.3390/f9100657.

Once again, thank you very much for your contribution to our work.
